

# Estimation of maize above-ground biomass based on stem-leaf separation strategy integrated with LiDAR and optical remote sensing data

Yaohui Zhu[1,2,3], Chunjiang Zhao[1,2,3], Hao Yang[2,3], Guijun Yang[2,3], Liang Han[2,4], Zhenhai Li[2,3], Haikuan Feng[2,3], Bo Xu[2,3], Jintao Wu[2,3] and Lei Lei[2,3]

[1] School of Information Science and Technology, Beijing Forestry University, Beijing, China
[2] Key Laboratory of Quantitative Remote Sensing in Agriculture of Ministry of Agriculture, Beijing Research Center for Information Technology in Agriculture, Beijing, China
[3] National Engineering Research Center for Information Technology in Agriculture, Beijing, China
[4] College of Architecture and Geomatics Engineering, Shanxi Datong University, Datong, China

Corresponding authors
Chunjiang Zhao, ait863@qq.com
Hao Yang, younghow@163.com

## ABSTRACT

Above-ground biomass (AGB) is an important indicator for effectively assessing crop growth and yield and, in addition, is an important ecological indicator for assessing the efficiency with which crops use light and store carbon in ecosystems. However, most existing methods using optical remote sensing to estimate AGB cannot observe structures below the maize canopy, which may lead to poor estimation accuracy. This paper proposes to use the stem-leaf separation strategy integrated with unmanned aerial vehicle LiDAR and multispectral image data to estimate the AGB in maize. First, the correlation matrix was used to screen optimal the LiDAR structural parameters (LSPs) and the spectral vegetation indices (SVIs). According to the screened indicators, the SVIs and the LSPs were subjected to multivariable linear regression (MLR) with the above-ground leaf biomass (AGLB) and above-ground stem biomass (AGSB), respectively. At the same time, all SVIs derived from multispectral data and all LSPs derived from LiDAR data were subjected to partial least squares regression (PLSR) with the AGLB and AGSB, respectively. Finally, the AGB was computed by adding the AGLB and the AGSB, and each was estimated by using the MLR and the PLSR methods, respectively. The results indicate a strong correlation between the estimated and field-observed AGB using the MLR method ($R^2$ = 0.82, RMSE = 79.80 g/m$^2$, NRMSE = 11.12%) and the PLSR method ($R^2$ = 0.86, RMSE = 72.28 g/m$^2$, NRMSE = 10.07%). The results indicate that PLSR more accurately estimates AGB than MLR, with $R^2$ increasing by 0.04, root mean square error (RMSE) decreasing by 7.52 g/m$^2$, and normalized root mean square error (NRMSE) decreasing by 1.05%. In addition, the AGB is more accurately estimated by combining LiDAR with multispectral data than LiDAR and multispectral data alone, with $R^2$ increasing by 0.13 and 0.30, respectively, RMSE decreasing by 22.89 and 54.92 g/m$^2$, respectively, and NRMSE decreasing by 4.46% and 7.65%, respectively. This study improves the prediction accuracy of AGB and

provides a new guideline for monitoring based on the fusion of multispectral and LiDAR data.

# INTRODUCTION

Maize is one of the main food crops today and is planted on a large scale worldwide. The timely and effective access to high-resolution spatial crop-development information provides important guidance for precision agricultural management, which allows the implementation of effective fertilization programs (*Cilia et al., 2014*; *Gracia-Romero et al., 2017*; *Samborski, Tremblay & Fallon, 2009*), irrigation measures (*Barker et al., 2018*; *Ma et al., 2018*; *Maresma, Lloveras & Martinez-Casasnovas, 2018*), and early production forecasts (*Elazab et al., 2016*; *Kitchen et al., 2003*; *Vergara-Diaz et al., 2016*). Crop above-ground biomass (AGB) is an important indicator for effectively assessing crop growth and yield, and also an important ecological indicator for assessing the efficiency of which crops use light and store carbon in ecosystems.

Accurate monitoring of crop AGB is an effective means to assess farmland productivity. Laboratory destructive methods to estimate biomass mainly involve inefficient and time-consuming manual sampling measurements, which is difficult to scale up for application to large areas. How to rapidly and accurately estimate the AGB of crops has always been a hot topic (*Li et al., 2015b*; *Wang et al., 2017*; *Zolkos, Goetz & Dubayah, 2013*). With its ability to acquire regional- and global-scale information, remote sensing technology has become an effective tool for estimating the AGB of crops over large regions. With the rapid development of remote sensing technology in recent years, multi-source remote sensing data acquired by unmanned aerial vehicle (UAV) fitted with spectral and LiDAR sensors have become widely applied. Unmanned aerial vehicles have been key to solve different problems in agriculture, which require high-precision crop data, such as crop pest detection (*Albetis et al., 2017*), crop yield estimation (*Zhou et al., 2017*), and crop variable measurement (*Bendig et al., 2015*). This approach allows high-efficiency and dynamic remote sensing monitoring of large areas in a convenient and nondestructive manner and with high throughput. Remote sensing from UAVs has thus become an important part of precision agriculture (*Liebisch et al., 2015*; *Marshall & Thenkabail, 2015*; *Yang et al., 2017*).

In spectroscopy, spectral sensors have been frequently cited as a rapid, nondestructive, and cost-effective tool for estimating agronomic parameters of different crops. With the rapid development of the UAV platform in recent years, the observation scale and timeliness of the spectrometer have rapidly improved (*Yang, Yang & Mo, 2018a*). Spectral measurements can be used to obtain appropriate spectral indicators from the visible and near-infrared spectral regions to estimate factors that describe crop-canopy growth such as biomass (*Liu et al., 2010*), leaf area index (*Potgieter et al., 2017*), and nitrogen content (*Fitzgerald, Rodriguez & O'Leary, 2010*). To evaluate the optimal method to

estimate crop biomass, many scholars have improved the accuracy of crop-biomass estimates by screening the optimal band or by combining the visible and near-infrared spectral bands (*Bendig et al., 2015*; *Fu et al., 2014*; *Gnyp et al., 2014*; *Kross et al., 2015*). Other studies have focused on combining crop-growth models with remote sensing data, and assimilating remote sensing data with uncertain input parameters of the crop-growth model, thereby improving the accuracy of crop-biomass predictions (*Jin et al., 2016*; *Machwitz et al., 2014*). As the spectral resolution of UAV multispectral cameras continues to improve and because the prices are reasonable, this platform has been widely used in vegetation phenotypic monitoring. However, the information obtained by the passive optical sensors mainly comes from the top of the vegetation, so very little information about the vertical structure of the vegetation can be obtained, which reduces the accuracy of the crop-biomass estimate (*Wang et al., 2017*).

LiDAR is a stable active remote sensing technology with strong penetration. It works with UAV platforms and provides accurate three-dimensional structural information of the vegetation canopy (*Wallace, Lucieer & Watson, 2014*). It provides rapid and nondestructive estimates of structural information, such as height, volume, leaf area index, and leaf area density of vegetation, which can resolve the problem of spectral saturation that occurs in optical remote sensing (*Cao et al., 2018*; *Du et al., 2016*; *Luo et al., 2018*; *Tagarakis et al., 2018*). Unlike manned aircraft, UAVs are being increasingly used to provide detailed, high-resolution imagery and associated digital elevation models (DEMs) for surface processes and geomorphological research (*James & Robson, 2014*). LiDAR has been widely used in many scenarios, especially for monitoring forest biomass (*Dubayah et al., 2010*; *Gonzalez de Tanago et al., 2018*; *Knapp, Fischer & Huth, 2018*; *Li et al., 2014, 2017*; *Nelson et al., 2017*; *Silva et al., 2017*; *Stovall et al., 2017*). However, LiDAR applications for crop biomass are fewer, and the penetration depth of LiDAR is limited because of the higher density of the crop canopy. A high-density LiDAR point cloud is a necessary condition for detecting crop-canopy density and increasing canopy penetration and can be used to directly quantify crop-structure parameters (*Christiansen et al., 2018*; *Eitel et al., 2014*; *Li et al., 2015a*).

In recent years, numerous studies have estimated vegetation biomass by using a combination that includes the three-dimensional structural information of the vegetation and the canopy spectral information (*Clark et al., 2011*; *Laurin et al., 2014*; *Swatantran et al., 2011*). This approach mainly involves (i) merging the LiDAR structural parameters (LSPs) extracted from airborne LiDAR data with the spectral vegetation indices (SVIs) extracted from satellite remote sensing images (*Li et al., 2015b*), (ii) merging the LSPs derived from airborne LiDAR data with the SVIs derived from hyperspectral imagery (*Luo et al., 2017*; *Wang et al., 2017*), (iii) merging the LSPs derived from ground-based LiDAR and the SVIs derived from hyperspectral data (*Tilly, Aasen & Bareth, 2015*), and (iv) merging the LSPs extracted from vehicle-based LiDAR data with the SVIs extracted from data acquired by active optical sensors (*Schaefer & Lamb, 2016*). However, previous studies have basically fused LSPs with spectral parameters at the data level to estimate vegetation biomass, and most of the research focuses on forestry. In contrast, few studies have focused on agriculture, and none have estimated the AGB of maize by separating the stem and leaf biomass.

This paper proposes a method to estimate the AGB of maize based on estimates of above-ground leaf biomass (AGLB) and above-ground stem biomass (AGSB) from multispectral data and LiDAR point cloud data, respectively, with both acquired from a UAV platform. By comparing the results of the two types of indicators with those of the AGB estimation model, we evaluate the value of integrating LiDAR data with multispectral data to estimate AGB. This study provides a good guideline for maize-field management based on using multispectral data to estimate leaf biomass and LiDAR point cloud data to estimate stem biomass. The main objectives of this study were: (1) to estimate the maize AGB based on a stem-leaf separation strategy integrated with LiDAR and optical remote sensing data; and (2) to compare the multivariable linear regression (MLR) and partial least squares regression (PLSR) models to determine which is most effective for estimating maize AGB.

## MATERIALS AND METHODS

### Study area

The study area was at the Xiaotangshan National Precision Agriculture Research and Demonstration Base of Beijing Academy of Agriculture and Forestry Sciences, Changping District, Beijing, China (40°00′–40°21′N, 116°34′–117°00′E). The site has an average altitude of 36 m, its total area is about 1.08 km$^2$, it has an average rainfall of 600 mm, and it is characterized by a typical north temperate semi-humid continental monsoon climate. The main type of crop grown in the area is maize, which is usually sown in late May, flowers in late July, and is harvested in mid-to-late September (*Li et al., 2015b*). The study area (15 × 15 m) contained 30 plots (3 × 1.25 m per plot) and was planted with the summer maize variety Jingdan 40. The field-management measures followed the standard practice for maize production (Fig. 1).

### Field measurement

The field measurements were made in the study area on August 28, 2018. The seedlings were treated, and the density of maize plants gradually decreased from north to south. To obtain the AGB of each sample, the data collectors first counted the number of maize plants in each plot. Next, they collected at random from each plot six plants with uniform growth, measured the plant height, and used the average height of the six plants as the measured height for the plot. All samples were then taken back to the laboratory to be oven dried at 80 °C until a constant weight was reached, and the dry weight (DW) of the maize leaves and stems were recorded separately. The average DW of the six maize plants in each plot was calculated and multiplied by the number N of corresponding plot plants to obtain the final DW. The DW was divided by the plot area, and the result converted to g/m$^2$. Finally, the AGB of each sample was measured.

### UAV multispectral data and digital image

The UAV multispectral data and digital image were collected from 11:00 to 13:00 during sunny and windless weather. The Parrot Sequoia multispectral camera (MicaSense Inc., Seattle, WA, USA) and the DJI FC6310 digital camera were simultaneously fit to the UAV

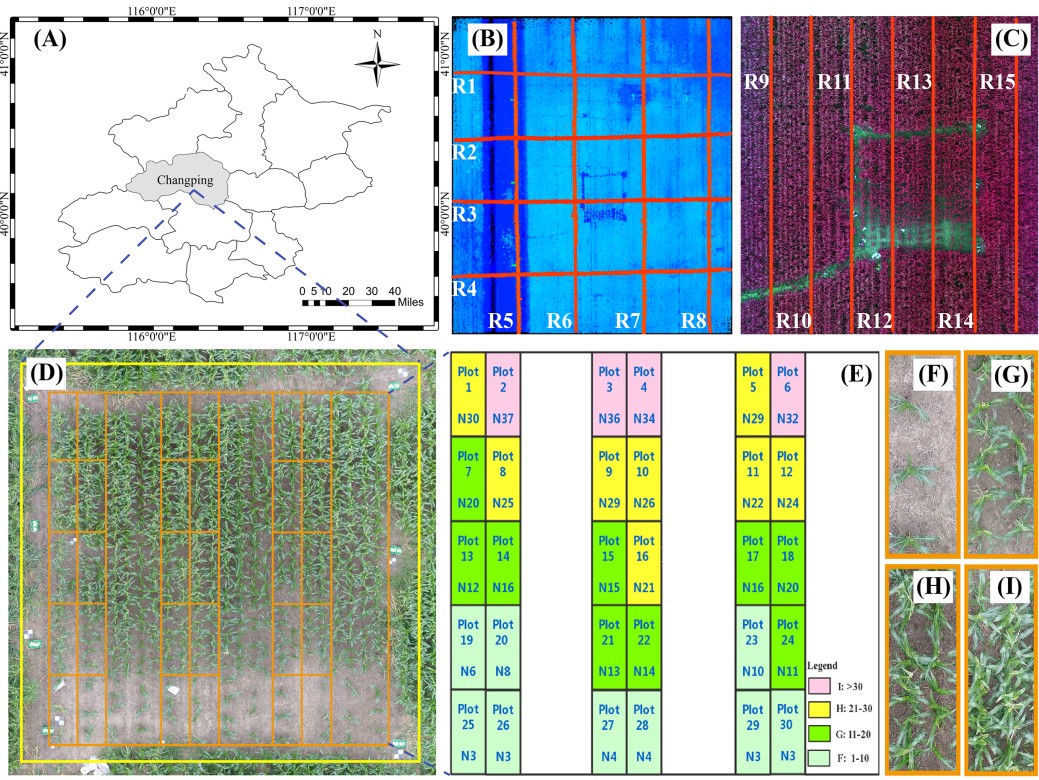

**Figure 1 Study area and experimental design.** (A) Location of study area in Beijing. (B) The red line in the figure is the flight path of the UAV with LiDAR. (C) The red line in the figure is the flight path of the UAV with multispectral capabilities. (D) The study area and field plot. (E) Experimental design. (F)–(I) represents the four levels of plant density. N is the number of plants. Plot represents the field plot.

platform of the DJI Phantom 4 Pro (SZ DJI Technology Co., Shenzhen, China) and are characterized by high precision, low weight, and ease of use. The acquired multispectral images contain four spectral channels: green (wavelength 550 nm, bandwidth 40 nm), red (660 nm, 40 nm), red edge (735 nm, 10 nm), and near infrared (790 nm, 40 nm). In addition, the DJI FC6310 digital camera is equipped with a 1-inch CMOS sensor with a resolution of 20 megapixels. The UAV flight height was set to 15 m, the speed to three m/s, the forward overlap to 80%, and the side overlap to 70%. The Parrot Sequoia multispectral camera was calibrated before and after the flight test by using a calibrated reflectance panel (MicaSense Inc., Seattle, WA, USA) to minimize error during image capture.

The pre-processing for multispectral image data, including mosaic, radiation calibration, and geometric correction, was handled by the data producer. The Agisoft PhotoScan Professional software (version 1.4.2, Agisoft LCC, St. Petersburg, Russia) was introduced into the multispectral camera's own calibration file to splice the multispectral images. The geometric correction and registration of multispectral images were done based on ground control points, and the spliced multispectral images in four spectral bands were then converted to reflectance by using the QUick Atmospheric Correction tool (Sensor type set to Generic/Unknown Sensor) in ENVI software (version 5.3.1, Esri Inc.,
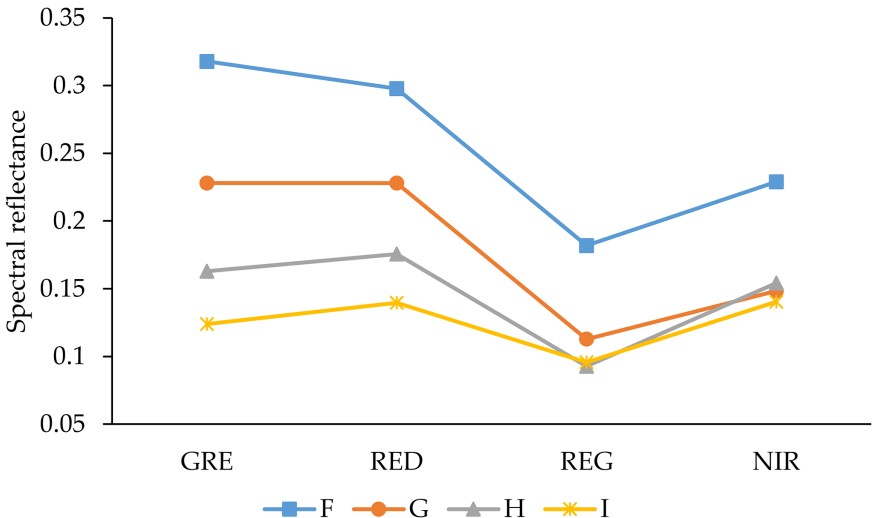

**Figure 2 Reflectance spectra of soils with different AGB.** GRE, RED, REG, and NIR represent the green band, red band, red edge band, and near-infrared band, respectively. F–I represents the density of plants.                                               

Redlands, CA, USA). The ground resolution and reprojection error of the spliced multispectral image are 1.33 cm/pix and 0.687 pix, respectively. The SVIs were calculated by using the bandmath tool to obtain the SVIs maps. Since these SVIs can respond to different ground objects, we used the bandmath tool to binarize the SVIs maps, and then separated plants from the soil background in these SVIs maps, where the spectral curves of the different AGB soils are shown in Fig. 2. In addition, ArcMap (version 10.3.1, Esri Inc., Redlands, CA, USA) was used to create the area of interest (AOI) with separated plant areas and to extract the average vegetation index for each plot.

Similar to the multispectral data pre-processing method, digital images with spatial position information were stitched by Agisoft PhotoScan Professional software, and geometric correction and registration of images based on ground control points were then exported to DEM images. The average plant height of each plot was extracted from ArcMap by using the AOI created in the previous step.

## UAV LiDAR point cloud data

The LiDAR data were synchronized with the multispectral image data-acquisition time, and the three-dimensional point cloud data of the study area plants were obtained by the UAV platform DJI M600 Pro (SZ DJI Technology Co., Shenzhen, China) equipped with a RIEGL VUX-1UAV (RIEGL Laser Measurement Systems, Ltd., Horn, Austria) laser scanner (the spot diameter was 0.0075 m, the average ground point spacing was 0.0239 m, the flying height was 15 m, the speed was three m/s, and the maximum scanning angle was 70°).

The RiPROCESS software (RIEGL Laser Measurement Systems, Ltd., Horn, Austria) was used for pre-processing, which included the analytical correction of multi-route LiDAR data. Post-processing of the LiDAR data was done by using the LiDAR360 software (version 2.0, GreenValle International, Ltd., Berkeley, CA, USA). First, the point cloud data

**Table 1 LiDAR point cloud parameter of different routes.**

| Direction | Route | Point cloud density (pts/m$^2$) | Spot diameter (cm) | Average ground point spacing (cm) | CHM resolution (cm) |
|---|---|---|---|---|---|
| EW | R1 | 112 | 0.75 | 2.39 | 1 |
| | R2 | 280 | | | |
| | R3 | 529 | | | |
| | R4 | 213 | | | |
| NS | R5 | 228 | | | |
| | R6 | 496 | | | |
| | R7 | 417 | | | |
| | R8 | 162 | | | |

**Note:**

EW indicates the east–west direction, NS indicates the north–south direction, and R1 to R8 are the various routes of the UAV-LiDAR.

were denoised. Next, the point cloud data of the study area were classified into ground points and non-ground points. The ground point was then normalized by the classified ground point cloud, which eliminates any influence of the terrain. Finally, the DEM and DSM with a high spatial resolution raster of $1 \times 1$ cm were obtained by using the terrain tool, and the CHM of the study area was calculated as the difference between the DSM and the DEM model. The vegetation height relative to the surface was obtained based on the CHM, and the standardized point cloud structure parameters were calculated from the LiDAR point cloud of the vegetation height. The detailed LiDAR point cloud parameters are shown in Table 1.

## Research route

Research route is shown in Fig. 3.

## Deriving metrics from multispectral data and LiDAR point cloud data

The physiological properties of vegetation imaged via remote sensing can be extracted by the vegetation indices. The reflectance of a given wavelength provides useful information about leaf-plant health. The vegetation indices are numbers that are computed from different wavelength reflectances by well-known equations that use the light reflectance of the plants in different bandwidths, especially the green, red, and near infrared (*Devia et al., 2019*). Vegetation indices have been widely used to estimate biomass by using empirical relationships with biomass (*Foody, Boyd & Cutler, 2003*). The more the vegetation grows, the more the red edge redshifts. When the vegetation grows under nutrient stress, the red edge blueshifts. In this study, the sensitivity of multispectral data to the physiological properties of vegetation and the multi-channel advantages of multispectral sensors combine to provide 10 vegetation indices commonly used in the literature to estimate crop biomass. These different vegetation indices were computed (Table 2) and include the Chlorophyll index green (CIgreen), Chlorophyll vegetation index (CVI), Enhanced vegetation index 2 (EVI2), Simple ratio greenness index (GI), Modified triangular vegetation index 2 (MTVI2), Normalized difference vegetation index (NDVI),

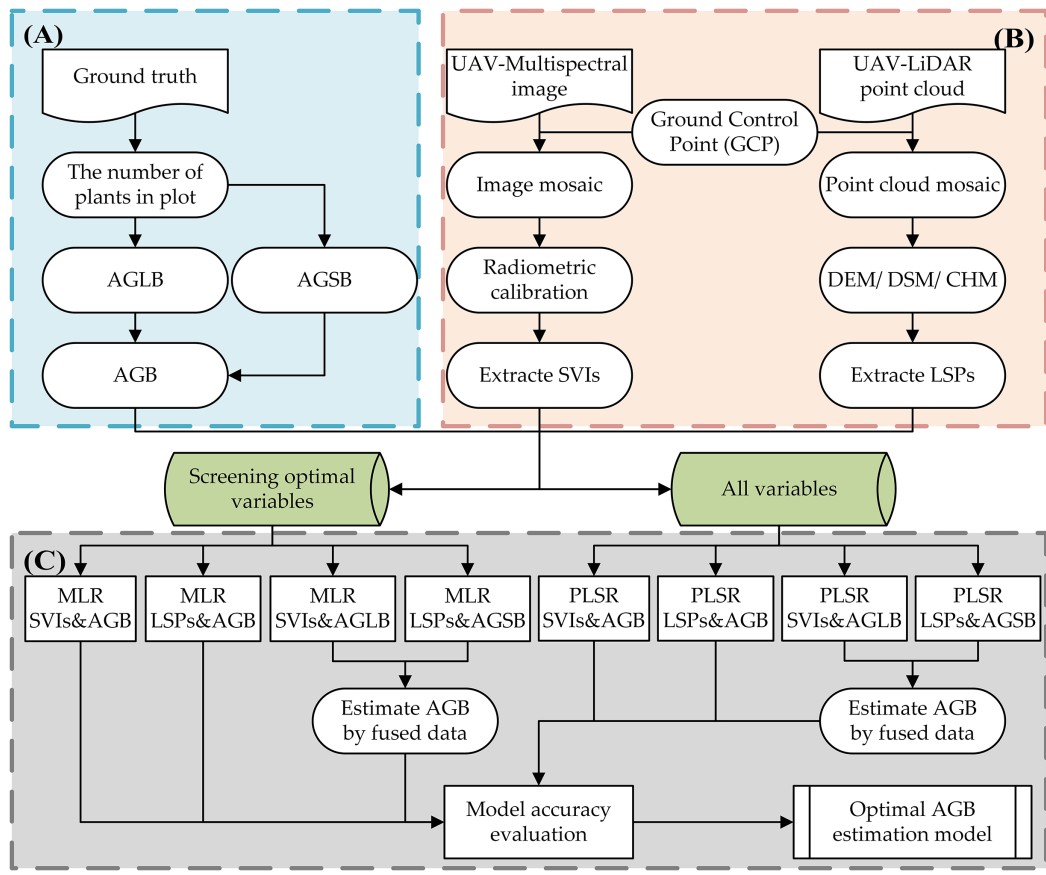

**Figure 3** **Research technology route.** (A) Field measurement data; (B) unmanned aerial vehicle LiDAR and multispectral image data preprocessing; (C) and estimation and verification of AGB.

**Table 2 Optical vegetation indices used in this study and citations for biomass.**

| Metric | Equations | References |
|---|---|---|
| CIgreen | $\rho_{NIR}/\rho_{green} - 1$ | *Gitelson et al. (2003)* |
| CVI | $\rho_{NIR} \times \rho_{red}/\rho_{green}^2$ | *Datt et al. (2003)* |
| EVI2 | $2.5(\rho_{NIR} - \rho_{red})/(\rho_{NIR} + 2.4\rho_{red} + 1)$ | *Jiang et al. (2008)* |
| GI | $\rho_{green}/\rho_{red}$ | *Smith et al. (1995)* |
| MTVI2 | $\dfrac{1.5\left(1.2\left(\rho_{NIR} - \rho_{green}\right) - 2.5\left(\rho_{red} - \rho_{green}\right)\right)}{\sqrt{\left(2\rho_{NIR} + 1\right)^2 - \left(6\rho_{NIR} - 5\sqrt{\rho_{red}}\right) - 0.5}}$ | *Haboudane et al. (2004)* |
| NDVI | $(\rho_{NIR} - \rho_{red})/(\rho_{NIR} + \rho_{red})$ | *Tucker et al. (1979)* |
| NGRDI | $\left(\rho_{green} - \rho_{red}\right)/\left(\rho_{green} + \rho_{red}\right)$ | *Zarco-Tejada et al. (2001)* |
| OSAVI | $(1 + Y)(\rho_{NIR} - \rho_{red})/(\rho_{NIR} + \rho_{red} + Y)(Y = 0.16)$ | *Rondeaux, Steven & Baret (1996)* |
| SAVI | $(1 + L)(\rho_{NIR} - \rho_{red})/(\rho_{NIR} + \rho_{red} + L)(L = 0.5)$ | *Huete (1988)* |
| SRVI | $\rho_{NIR}/\rho_{red}$ | *Birth & McVey (1968)* |

**Note:**
In this table, $\rho$ is the abbreviation for reflectance.

**Table 3 LiDAR-derived metrics for estimating biomass parameters.**

| Metric | Equations | Description |
|---|---|---|
| H_max | $H\_max = max(H_i), 1 \le i \le N$ | Maximum height of plants |
| H_mean | $H\_mean = \dfrac{1}{N}\sum_{i=1}^{N} H_i$ | Mean height of plants |
| H_sd | $H\_sd = \sqrt{\dfrac{1}{N-1}\sum_{i=1}^{N}(H_i - H\_mean)^2}$ | Standard deviation of plants height |
| H_cv | $H\_cv = H\_sd/H\_mean$ | Variation coefficient of plants height |

**Note:**
In the table, $H_i$ is the height of maize plants.

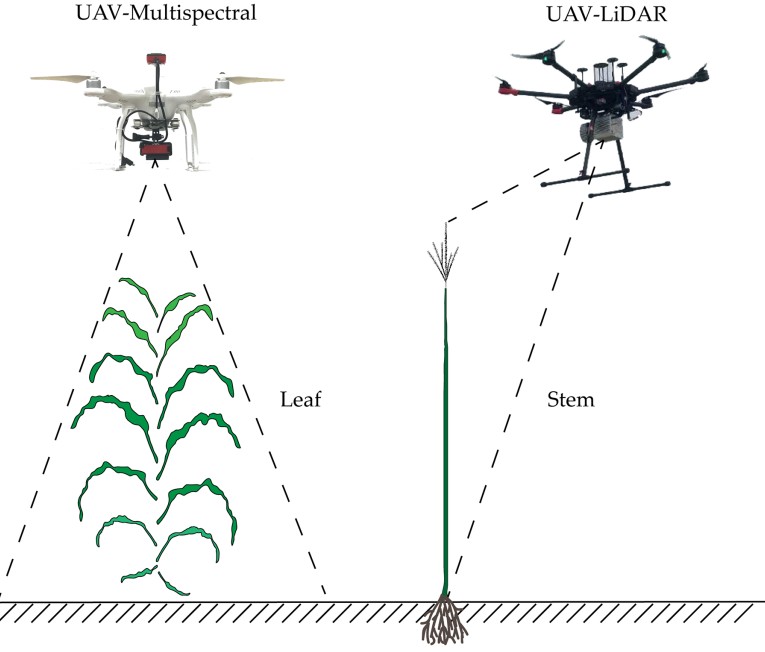

UAV-Multispectral          UAV-LiDAR

Leaf          Stem

**Figure 4 Schematic diagram of estimation of maize AGB based on stem-leaf separation strategy integrated with LiDAR and multispectral data.**

Normalized green red difference index (NGRDI), Optimized soil adjusted vegetation index (OSAVI), Soil adjusted vegetation index (SAVI), and Simple ratio vegetation index (SRVI).

In addition, based on the point cloud structural parameters used in previous studies that focused on estimating crop biomass, we selected four common LSPs: average height (H_mean), maximum height (H_max), height standard deviation (H_sd), and height coefficient of variation (H_cv). Table 3 gives detailed definitions and explanations of these four point cloud structural parameters.

## Estimation and verification of above-ground biomass

We propose herein a method for estimating the AGB of maize that uses estimated AGLB and AGSB obtained from multispectral data and LiDAR point cloud data (Fig. 4). As opposed to previous methods of predicting AGB by LiDAR and optical remote sensing data in their respective or combined forms, we divided the measurement of maize AGB

into two parts: AGLB and AGSB. The method combines the respective advantages of multispectral data and LiDAR point cloud data to measure AGLB and AGSB, respectively. The AGLB of maize was measured by using multispectral data sensitive to the vegetation canopy; the AGSB of maize was measured by using LiDAR point cloud data sensitive to vegetation structure. Finally, the maize AGB was obtained by accumulating the AGLB and AGSB measured in each plot.

In this study, MLR and PLSR were used for maize AGB measurements. First, the correlation between SVIs (CIgreen, CVI, EVI2, GI, MTVI2, NDVI, NGRDI, OSAVI, SAVI, SRVI), LSPs (H_max, H_mean, H_sd, H_cv), and field-measured AGLB and AGSB was used to determine the relationship between the two types of indicators and the above-ground leaf and stem biomass, and the optimal SVIs and LSPs were screened. Next, following previous studies (*Bendig et al., 2015*), we used MLR methods to estimate crop biomass. Based on the optimized LSPs and SVIs for estimating biomass, a MLR model was then constructed with AGLB and AGSB. We then use all SVIs and LSPs to estimate maize AGB based on the PLSR method. To solve the over-fitting problem of the model, the cross-validation method was used to determine the appropriate number of important factors in the PLSR model. Finally, the optimal method was selected by comparing the effectiveness of MLR and PLSR in estimating corn AGB. In addition, we evaluated the accuracy of the AGB estimation model based on the stem-leaf separation strategy by using LiDAR and multispectral metrics.

## Statistical analysis

In this study, the correlation coefficient between the predicted biomass and the measured biomass was evaluated by using $R^2$, which measures the relationship between two datasets and describes the proportion of the total variance in the measured data that can be explained by the model (Eq. (1)). $R^2$ ranges between 0 and 1 with higher values indicating better simulations. The discrepancy between the predicted and measured values of AGB is evaluated based on the root mean square error (RMSE) and normalized root mean square error (NRMSE). The RMSE serves to measure the extent to which the observed value deviates from the measured value and is very sensitive to the error response (Eq. (2)). The smaller the RMSE, the more accurate the measurement. However, it was difficult to estimate the actual gap between the predicted value and the measured value by RMSE because no specific measurement standard is available. The NRMSE helps to compare datasets or models that use different scales and is usually expressed as a percent, where a lower percent indicates a smaller residual variance and, typically, the model accuracy is excellent if NRMSE is less than 10%, good if NRMSE is between 10% and 20%, fair if NRMSE is between 20% and 30%, and poor if NRMSE exceeds 30% (*Ahmadi et al., 2015*) (Eq. (3)). Therefore, the use of RMSE and NRMSE as indicators for evaluation better reflects the actual accuracy of the model. Meanwhile, one-way analysis of variance (ANOVA) was used to test whether significant differences existed between above-ground biomass at different density levels ($0.01 \leq p \leq 0.05$ indicates a significant difference; $p < 0.01$ means an extremely significant difference).

**Table 4 Basic statistics of the plant height measurements.**

| Data | Mean (m) | Maximum (m) | Minimum (m) | SD (m²) | CV (%) |
|------|----------|-------------|-------------|---------|--------|
| Digital | 1.86 | 2.29 | 1.06 | 0.39 | 0.21 |
| LiDAR | 2.23 | 2.51 | 2.02 | 0.13 | 0.06 |
| Ground-truth | 2.18 | 2.42 | 1.97 | 0.13 | 0.06 |

$$R^2 = 1 - \frac{\sum_{t=1}^{n} (\hat{y}_t - \bar{y})^2}{\sum_{t=1}^{n} (y_t - \bar{y})^2} \tag{1}$$

$$\text{RMSE} = \sqrt{\frac{\sum_{t=1}^{n} (\hat{y}_t - y_t)^2}{n}} \tag{2}$$

$$\text{NRMSE} = \frac{\text{RMSE}}{y_{\max} - y_{\min}} \tag{3}$$

In the above formulas, $n$ is the number of samples, $\hat{y}_t$ is the values calculated with models, $y_t$ is the measured value; $\bar{y}$ is the average value, $y_{\max}$ is the maximum value, and $y_{\min}$ is the minimum value.

# RESULTS

## Plant height extracted from digital image and LiDAR data

The average plant height of each plot was extracted from ArcMap by using the AOI created in the previous step. Subsequently, plant heights of 30 plots were obtained from digital images and LiDAR point cloud data, respectively. Table 4 shows the average, maximum, minimum, and coefficient of variation with plant heights derived from the image, LiDAR point cloud, and ground measured. The spot diameter and the average ground point spacing of the LiDAR point cloud were 0.0075 and 0.0239 m, respectively.

## Screening for optimal variables

In this study, we constructed the correlation matrix for SVIs and AGB and AGLB and found a significant correlation between different SVIs and AGB and AGLB. Compared with the AGB, the correlation between the AGLB and the vegetation index is higher because the stem of maize cannot be observed in the multispectral image, so the SVIs are more sensitive to the AGLB, as shown in Fig. 5A. Because of the high correlation between the SVIs, the multivariate collinearity between indices may be problematic. Therefore, two vegetation indices, NGRDI and SRVI, with the highest correlation coefficient with the corresponding biomass (NGRDI-AGB/AGLB: 0.75/0.85, SRVI-AGB/AGLB: 0.72/0.83) were selected as multivariate variables to prevent overfitting and to reduce the complexity of the model. Compared with other indices, the NGRDI is very effective for monitoring AGB (*Elazab et al., 2016*), thereby improving the accuracy of models that estimate AGB. Fig. 5A shows that the AGLB is more correlated with SVIs than AGB, which further confirms that the SVIs are more sensitive to the AGLB.

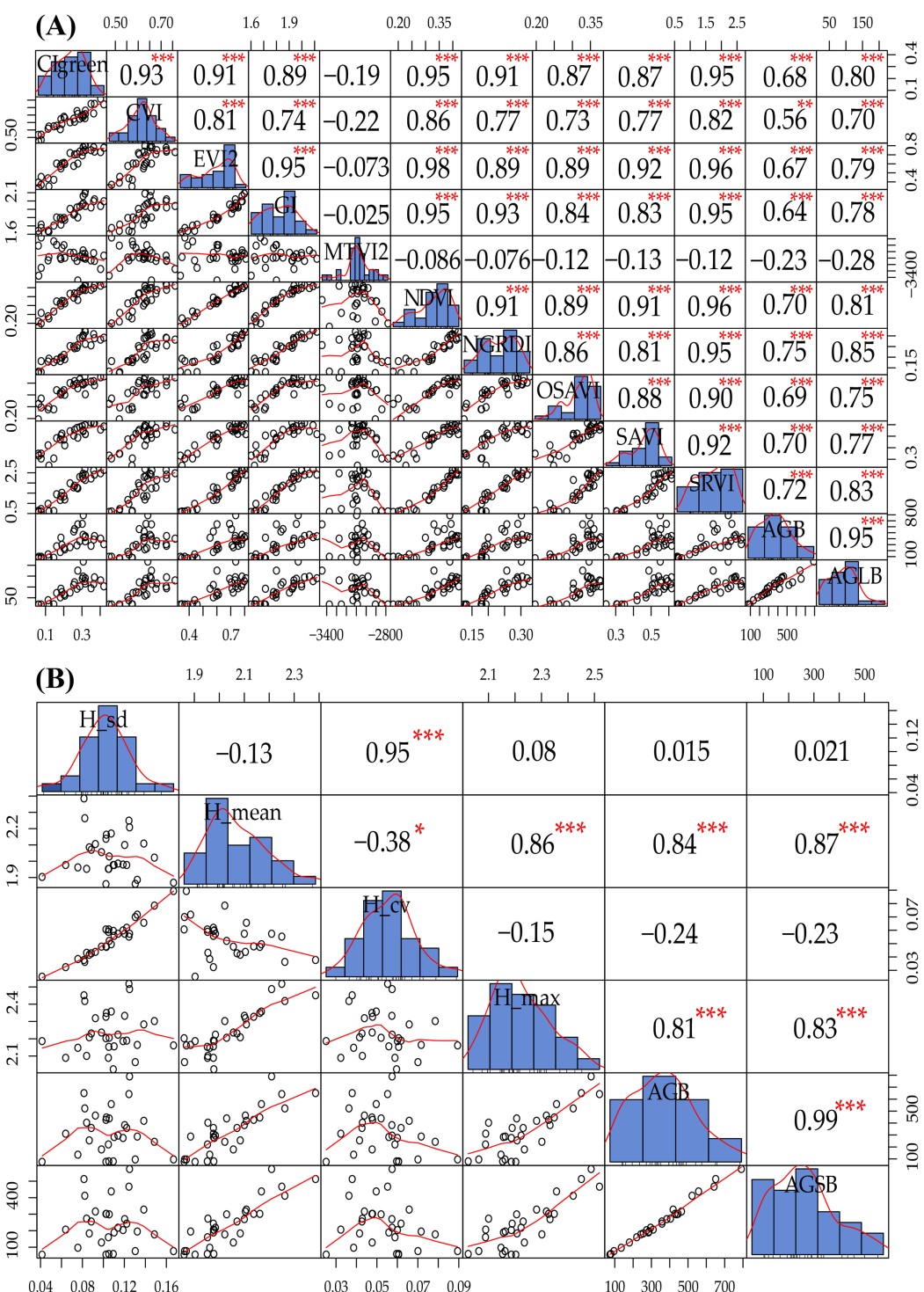

**Figure 5 (A) Data exploration of SVIs and AGB and AGLB; (B) data exploration of LSPs and AGB and AGSB.** The distribution of the variable itself is given in the diagonal area; the scatter plot and the curve fit between the two attributes are given at the lower left of the diagonal; the upper-right digit of the diagonal indicates the relationship between the two attributes; the asterisk * indicates the degree of significance between attributes.

Similarly, we constructed a correlation matrix for LSPs, AGB, and AGSB and then found that different LSPs have different sensitivities to AGB and AGSB, as shown in Fig. 5B. Because H_mean and H_max are significantly correlated with AGB and AGSB (H_mean-AGB/AGSB: 0.84/0.87, H_max-AGB/AGSB: 0.81/0.83), we select these two structural parameters as multivariate variables to construct the MLR model. Figure 5B shows that the AGSB is more correlated with LSPs than AGB, which further confirms that the LSPs are more sensitive to the AGSB.

## Estimate of maize biomass from multispectral data

In this study, we constructed a MLR model by screening multiple-regression variables (NGRDI and SRVI). The AGB was measured from 20 (2/3) samples as the modeling set, and the remaining 10 (1/3) samples were used as validation sets to evaluate the accuracy of the AGB estimate based on multispectral data. The results show that the variance of 0.67 is explained by the modeling set data (RMSE = 119.03 g/m$^2$, NRMSE = 16.59%) in the AGB measurement, and the variance of 0.18 is explained by the validation set data (RMSE = 136.76 g/m$^2$, NRMSE = 28.51%) in the AGB measurement. Figure 6A shows a scatterplot of field-observed biomass versus estimated biomass using two vegetation indices ($R^2$ = 0.56, RMSE = 125.38 g/m$^2$, NRMSE = 17.47%), and introduce 95% confidence level (*Wang et al., 2018a*). In addition, we constructed a PLSR model by all SVIs, with 20 (2/3) samples as the modeling set; the remaining 10 (1/3) samples were used as validation sets to evaluate the accuracy with which the AGB is estimated based on multispectral data. At the same time, two important factors in the PLSR model were determined by using the cross-validation method. The results show that the variance of 0.67 is explained by the modeling set data (RMSE = 118.40 g/m$^2$, NRMSE = 16.49%) in the AGB measurement, and the variance of 0.32 is explained by the validation set data (RMSE = 143.18 g/m$^2$, NRMSE = 29.84%) in the AGB measurement. Figure 6B shows a scatterplot of field-observed biomass versus estimated biomass using all vegetation indices ($R^2$ = 0.56, RMSE = 127.20 g/m$^2$, NRMSE = 17.72%).

The results show a low correlation between field-observed biomass and estimated biomass, which is tentatively attributed to the stem of maize not appearing in the multispectral images and the maize AGLB only accounting for about 25% of the AGB. Meanwhile, no significant difference appears in the accuracy of the AGB-SVIs estimation model constructed by the MLR and PLSR methods. Thus, the ability to predict AGB by using multispectral data is limited.

## Estimate of maize biomass from LiDAR data

In this study, we constructed a MLR model by screening multiple-regression variables (H_mean and H_max) and measured AGB, with 20 (2/3) samples as the modeling set and the remaining 10 (1/3) samples serving as validation sets to evaluate the accuracy with which the AGB is estimated based on LiDAR data. The results show that the variance of 0.77 is explained by the modeling set data (RMSE = 98.81 g/m$^2$, NRMSE = 13.77%) in the AGB measurement, and the variance of 0.53 is explained by the validation set data

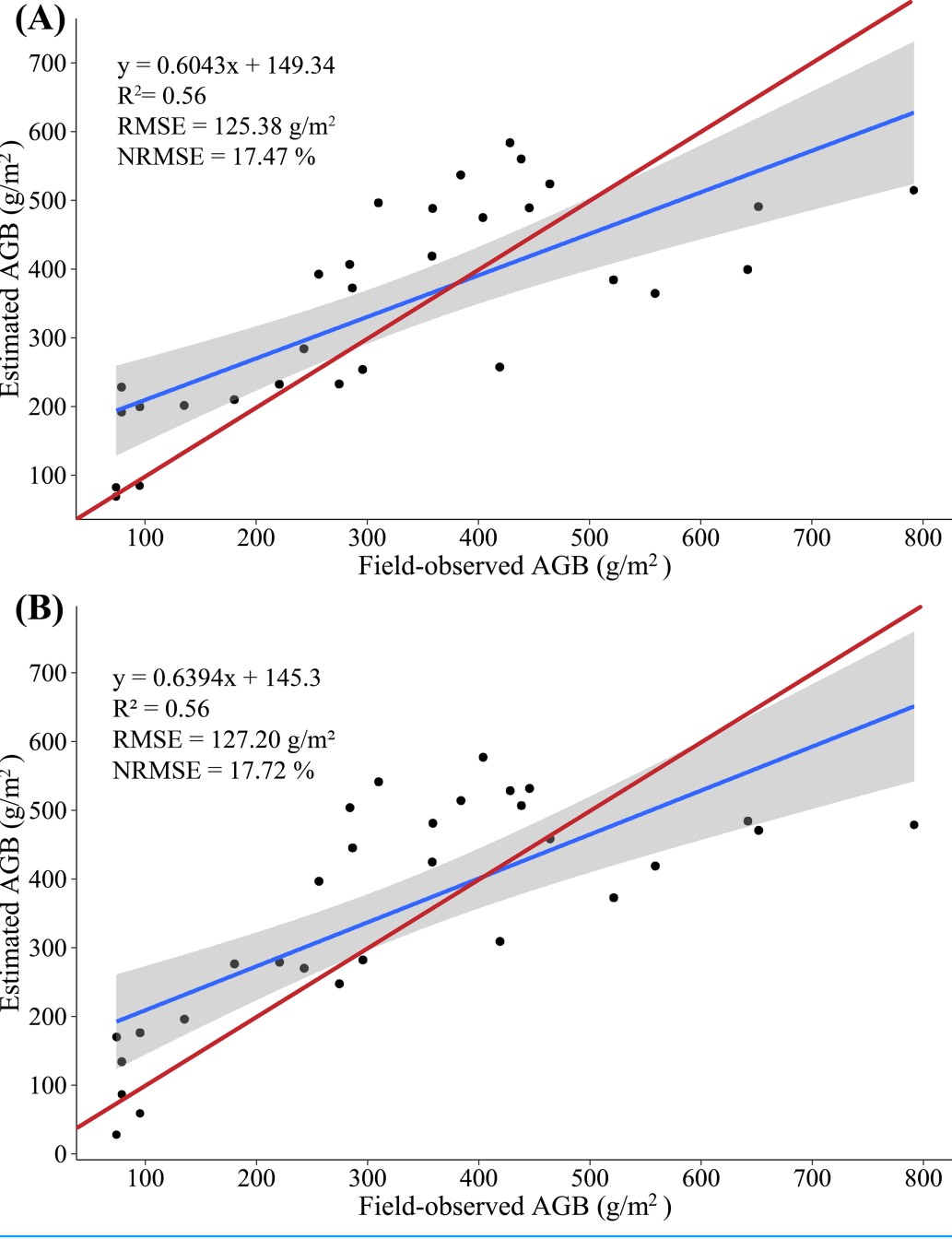

**Figure 6 (A) Scatterplot of field-observed AGB versus AGB estimated by using two SVIs (MLR);
(B) scatterplot of field-observed AGB versus AGB estimated by using all SVIs (PLSR).** The red line
represents the 1:1 line.                               

(RMSE = 91.04 g/m$^2$, NRMSE = 18.98%) in the AGB measurement. Figure 7A shows a
scatterplot of the field-observed AGB versus the AGB estimated by using two LSPs
($R^2$ = 0.73, RMSE = 96.29 g/m$^2$, NRMSE = 13.41%). In addition, we constructed a
PLSR model by all LSPs, with 20 (2/3) samples as the modeling set and the remaining
10 (1/3) samples serving as validation sets to evaluate the accuracy with which the AGB is

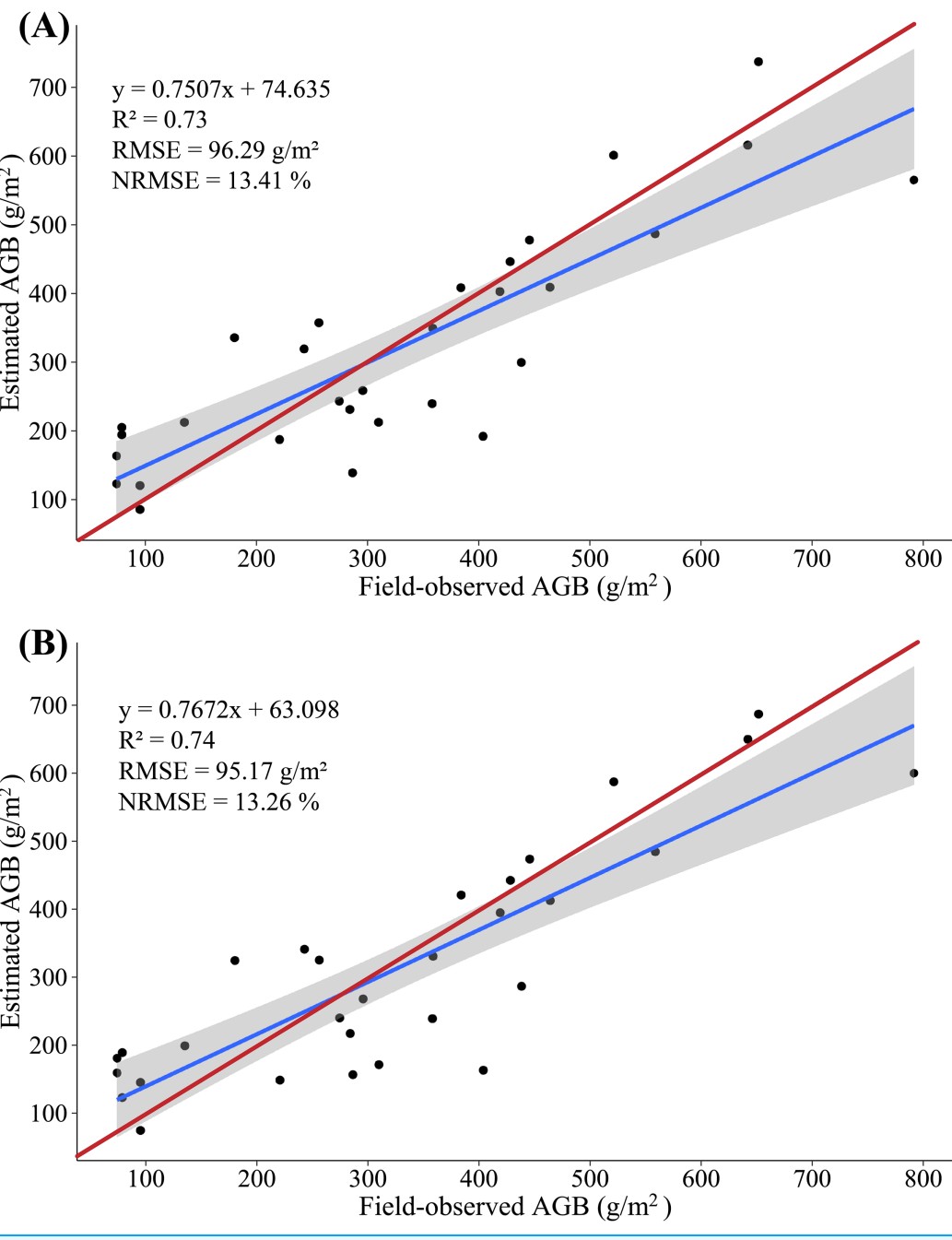

**Figure 7 (A) Scatterplot of field-observed AGB versus AGB estimated by using two LSPs (MLR); (B) scatterplot of field-observed AGB versus AGB estimated by using all LSPs (PLSR).** The red line represents the 1:1 line.

estimated based on LiDAR data. At the same time, two important factors in the PLSR model are determined by using the cross-validation method. The results show that the variance of 0.76 is explained by the modeling set data (RMSE = 99.92 g/m$^2$, NRMSE = 13.92%) in the AGB measurement, and the variance of 0.66 is explained by the validation set data (RMSE = 84.85 g/m$^2$, NRMSE = 17.68%) in the AGB measurement. Figure 7B

shows a scatterplot of the field-observed AGB versus the AGB estimated by using two LSPs ($R^2$ = 0.74, RMSE = 95.17 g/m$^2$, NRMSE = 13.26%).

The accuracy of AGB estimate based on LiDAR data clearly exceeds that of the AGB estimate based on multispectral data: $R^2$ increases by 0.17–0.18, RMSE decreases by 29.08–32.03 g/m$^2$, NRMSE decreases by 4.36–4.46%, and the estimated biomass is consistent with the field-observed biomass. These results further confirm a strong correlation between the LSPs and the AGB. The MLR and PLSR model constructed from the LSPs can better estimate the AGB of maize, which is consistent with the results of previous studies (*Wang et al., 2017*).

## Estimate of maize AGB from fusion of multispectral and LiDAR data compared with multispectral and LiDAR data only

In this study, because of the sensitivity of multispectral data to AGLB and of LiDAR data to AGSB, we propose a method for estimating maize biomass in which the AGLB and AGSB are estimated based on the multispectral data and LiDAR data, respectively. According to the screened indicators, the SVIs derived from multispectral data and the LSPs derived from LiDAR data were subjected to MLR with the AGLB and AGSB, respectively. At the same time, all SVIs derived from multispectral data and all LSPs derived from LiDAR data were subjected to PLSR with the AGLB and AGSB, respectively. Similarly, 20 (2/3) samples were used as the modeling set, and the remaining 10 (1/3) samples were used as the verification set (Figs. 8 and 9).

The results show that the MLR model constructed by SVIs and AGLB with the variance of 0.77 is explained by the modeling set data (RMSE = 27.39 g/m$^2$, NRMSE = 13.84%), and the variance of 0.57 is explained by the validation set data (RMSE = 26.15 g/m$^2$, NRMSE = 25.78%). Figure 8A shows a scatterplot of the field-observed AGLB versus the AGLB estimated by using two vegetation indices ($R^2$ = 0.72, RMSE = 26.98 g/m$^2$, NRMSE = 13.63%). Next, the results show that the MLR model constructed by the LSPs and AGSB, with a variance of 0.81, is explained by the modeling set data (RMSE = 66.46 g/m$^2$, NRMSE = 12.79%), and the variance of 0.61 is explained by the validation set data (RMSE = 64.92 g/m$^2$, NRMSE = 17.16%). Figure 8B shows a scatterplot of the field-observed AGSB versus the AGSB estimated by using two LSPs ($R^2$ = 0.77, RMSE = 65.95 g/m$^2$, NRMSE = 12.68%).

In addition, two important factors in the PLSR model are determined by using the cross-validation method. The results show that the PLSR model constructed by SVIs and AGLB with the variance of 0.78 is explained by the modeling set data (RMSE = 26.55 g/m$^2$, NRMSE = 13.41%), and the variance of 0.70 is explained by the validation set data (RMSE = 29.46 g/m$^2$, NRMSE = 29.04%). Figure 9A shows a scatterplot of the field-observed AGLB versus the AGLB estimated by using all vegetation indices ($R^2$ = 0.73, RMSE = 27.56 g/m$^2$, NRMSE = 13.92%). Next, the results show that the PLSR model constructed by the LSPs and AGSB, with a variance of 0.80, is explained by the modeling set data (RMSE = 68.23 g/m$^2$, NRMSE = 13.12%), and the variance of 0.75 is explained by the validation set data (RMSE = 59.02 g/m$^2$, NRMSE = 15.60%). Figure 9B shows a

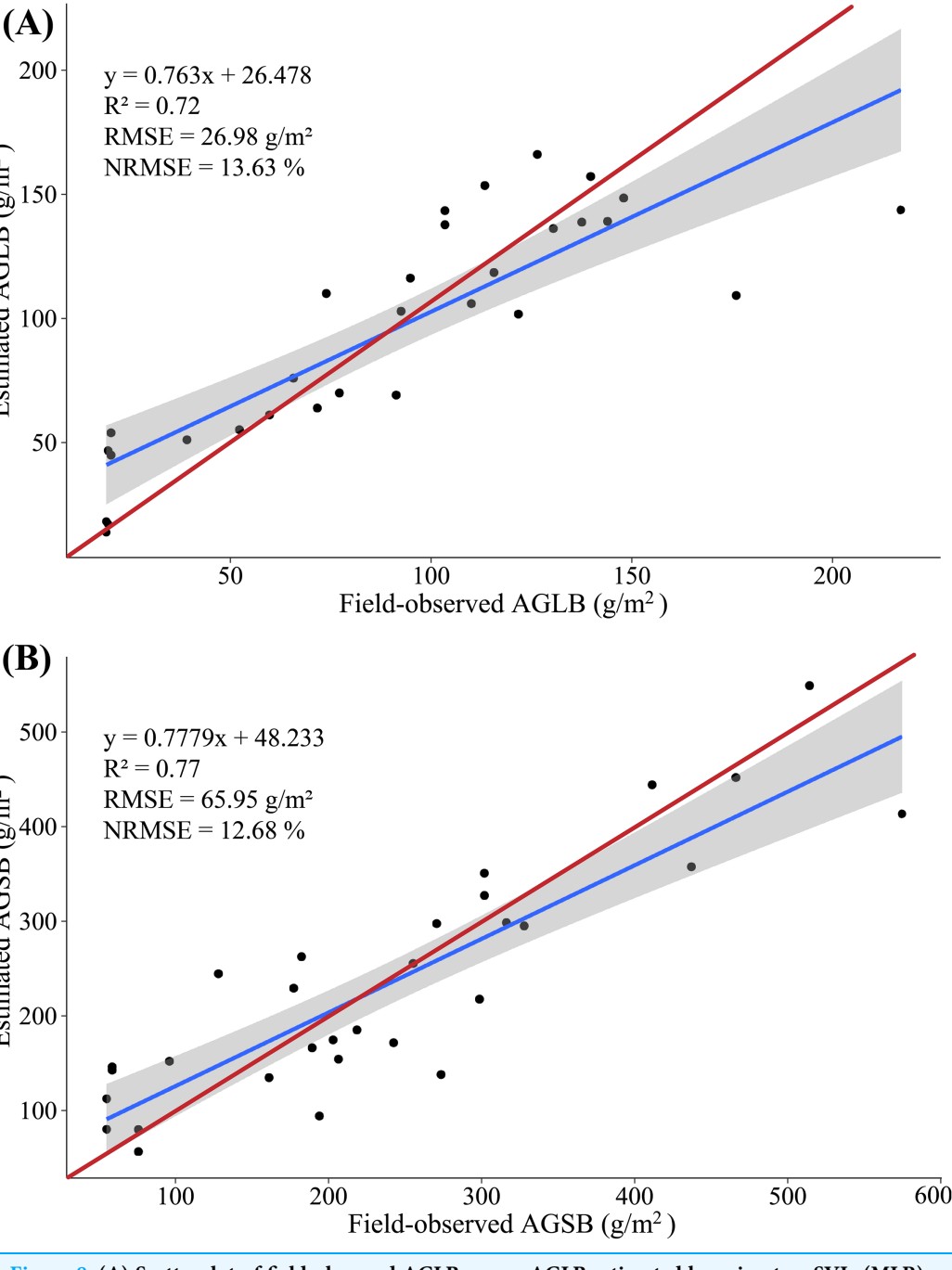

**Figure 8** **(A) Scatterplot of field-observed AGLB versus AGLB estimated by using two SVIs (MLR);** **(B) scatterplot of field-observed AGSB versus AGSB estimated by using two LSPs (MLR).** The red line represents the 1:1 line.                                                               

scatterplot of the field-observed AGSB versus the AGSB estimated by using all LSPs ($R^2 = 0.78$, RMSE = 65.31 g/m$^2$, NRMSE = 12.57%).

Overall, compared with the AGB, the multispectral data estimate of the AGLB leads to an increase in $R^2$ of 0.16–0.17, which means that the multispectral data are more sensitive to the AGLB. In addition, the LiDAR-data estimate of the AGSB leads to an increase in

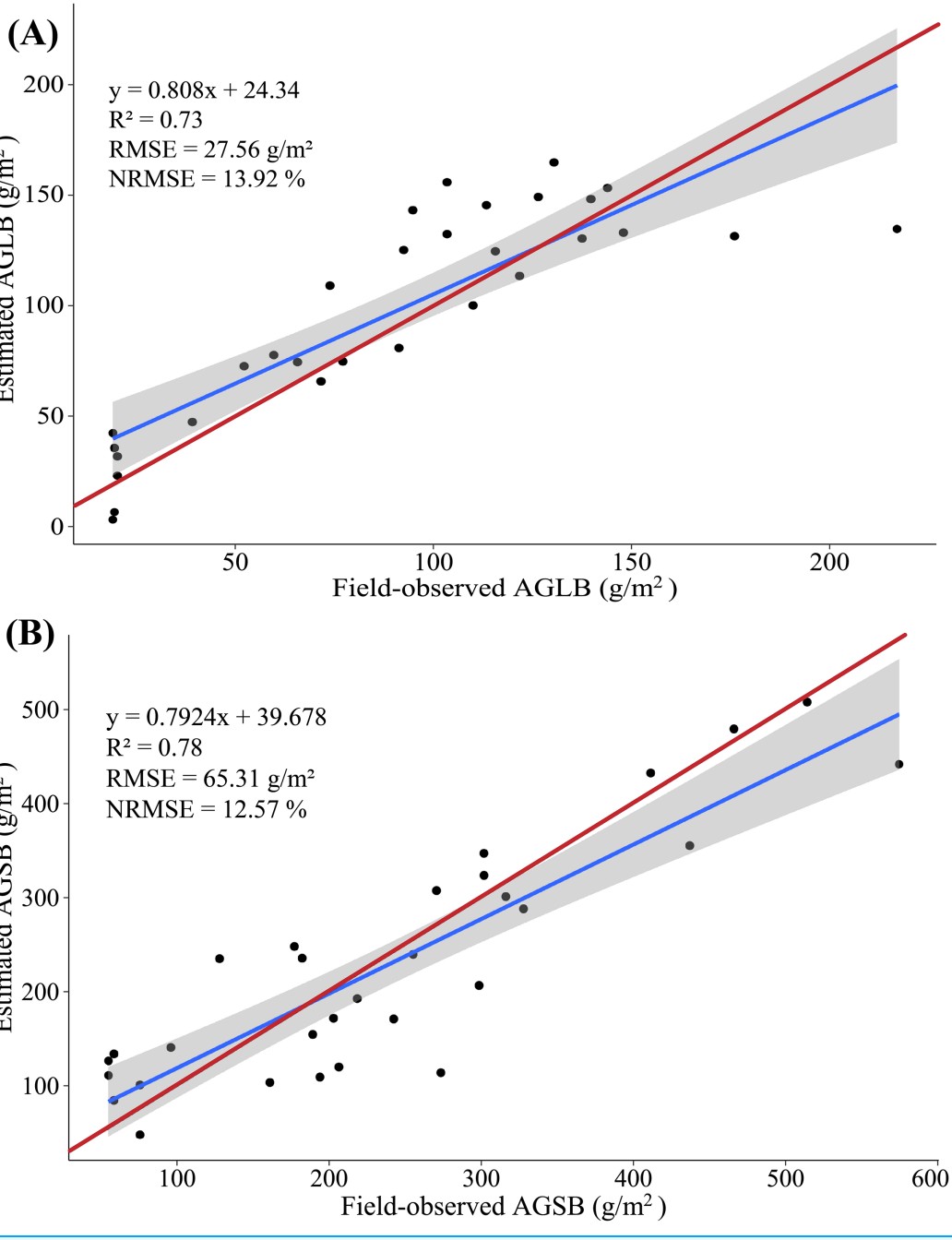

**Figure 9 (A) Scatterplot of field-observed AGLB versus AGLB estimated by using all SVIs (PLSR); (B) scatterplot of field-observed AGSB versus AGSB estimated by using all LSPs (PLSR).** The red line represents the 1:1 line.

$R^2$ of 0.04–0.05, which means that the LiDAR data are more sensitive to the AGSB. We then estimated the AGB by accumulating the estimated AGLB and the estimated AGSB using MLR and PLSR methods, respectively, and a linear regression was constructed by using the field-observed AGB and the estimated AGB. The results show that the variance of 0.82 is explained by the synergistic use of multispectral and LiDAR data

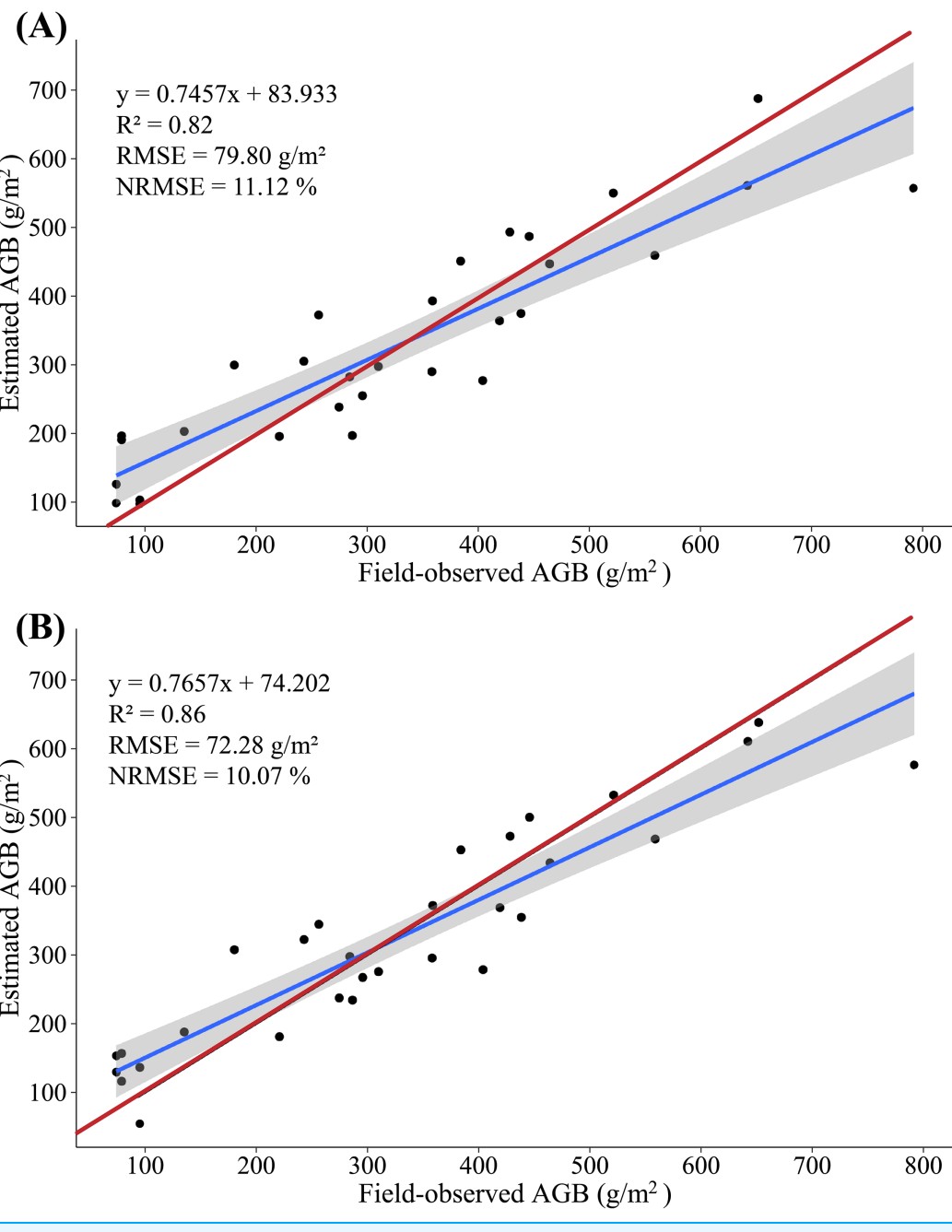

**Figure 10** (A) Scatterplot of field-observed AGB versus AGB estimated by merging multispectral and LiDAR data (MLR); (B) scatterplot of field-observed AGB versus AGB estimated by merging multispectral and LiDAR data (PLSR). The red line represents the 1:1 line.

(RMSE = 79.80 g/m$^2$, NRMSE = 11.12%) in the AGB measurement using MLR method (Fig. 10A); and the variance of 0.86 is explained by the synergistic use of multispectral and LiDAR data (RMSE = 72.28 g/m$^2$, NRMSE = 10.07%) in the AGB measurement using PLSR method (Fig. 10B).

**Table 5 Modeling statistics between the estimated and the measured AGB.**

| Models | Different combinations | Verification | | |
|--------|------------------------|--------------|---|---|
| | | $R^2$ | RMSE (g/m$^2$) | NRMSE (%) |
| MLR | AGB-SVI | 0.56 | 125.38 | 17.47 |
| | AGB-LSP | 0.73 | 96.29 | 13.41 |
| | AGB-SVI+LSP | 0.82 | 79.80 | 11.12 |
| PLSR | AGB-SVI | 0.56 | 127.20 | 17.72 |
| | AGB-LSP | 0.74 | 95.17 | 13.26 |
| | AGB-SVI+LSP | 0.86 | 72.28 | 10.07 |

**Note:**
In the table, AGB-SVI, AGB-LSP, and AGB-SVI+LSP represent three different combinations for MLR method. AGB-SVI + LSP represents the estimation of AGB based on stem-leaf separation strategy with LiDAR and multispectral data.

The study found that PLSR is more accurate for AGB estimation than MLR, with $R^2$ increasing by 0.04, RMSE decreasing by 7.52 g/m$^2$, and NRMSE decreasing by 1.05%. Instead of using the SVIs and LSPs separately to estimate AGB, this study estimates the AGB by using the AGLB and AGSB of maize estimated from multispectral and LiDAR data, respectively. By merging the estimates based on LiDAR and multispectral data, the method improves the accuracy with which the maize AGB is estimated. This study thus gives good results that indicate a high potential for estimating maize AGB based on stem-leaf separation strategy (Table 5).

To illustrate the difference in AGB between different densities, the results of the ANOVA analysis show that the *p*-value is less than 0.001, which indicates that there were statistical differences between the groups at different density levels. We therefore do a multiple comparative analysis in the discussion.

## DISCUSSION

Accurate monitoring of maize AGB can provide valuable guidance for agricultural production. In this study, MLR and PLSR were used for maize AGB measurements. To avoid the multi-collinearity and over-fitting problems of the MLR model, all the predictors are screened in advance to obtain the optimal variables. The obvious advantage of the MLR method is that it is highly interpretable, and its standardized partial regression coefficient determines the strength of independent variables versus dependent variables (*Han et al., 2019*). Compared with the traditional MLR model, however, the PLSR model concentrates on the characteristics of principal component regression (PCR) and MLR methods in the modeling process (*Geladi & Kowalski, 1986*). Therefore, the PLSR model can allow regression modeling under the condition that the independent variables have multiple collinearities to find latent structures in a large number of variables by reducing the number of variables to a few noncorrelated principal components (*Cho et al., 2007*; *Næsset, Bollandsås & Gobakken, 2005*). Although the PLSR model has higher prediction accuracy, both methods achieve acceptable accuracy. To obtain the optimal training model, sufficient samples are necessary. Considering that the actual sample size of this study is relatively small, no attempt was made to use these methods, such as artificial

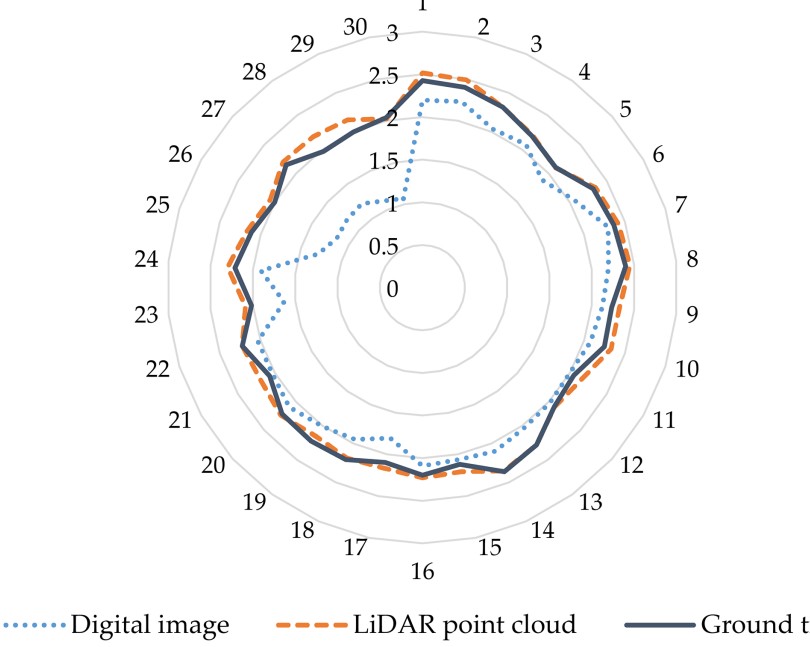

······· Digital image      ----- LiDAR point cloud      —— Ground truth

**Figure 11 Comparison of accuracy of maize plant height extracted from digital and LiDAR data.** The blue dashed circle in the figure represents the height parameters of the extracted digital image contain some outliers in the lower planting density area. The outer number represents the field plot, and the inner number represents the plant height.               

neural network (*Vahedi, 2016*; *Xie et al., 2009*; *Yang et al., 2018b*) or support vector machine (SVM) (*Clevers et al., 2007*; *Marabel & Alvarez-Taboada, 2013*). In fact, we emphasize herein the idea of stem-leaf separation modeling. The application of this idea in the above machine learning model will be realized after the sample is expanded in future experiments.

   Plant height is an important morphological and phenotypic indicator that directly indicates the overall growth of plants and predicts crop biomass and yield. Therefore, obtaining high-precision vegetation height is an important factor for accurately estimating vegetation biomass (*Wang et al., 2018b*). In previous studies, many scholars used digital and LiDAR data to estimate vegetation height (*Jensen & Mathews, 2016*; *Madec et al., 2017*; *Wallace et al., 2016*). In the present study, the plant height was determined based on digital and LiDAR point cloud data, and the results were verified by comparison with the measured plant height on artificial ground. The results show that, compared with the LiDAR point cloud data, the height parameters of the extracted digital image contain some outliers, which differ significantly from the values measured on the ground. However, the height parameters derived from the canopy three-dimensional dense point cloud data acquired by the LiDAR sensor are strongly consistent with the values measured on the ground, and the accuracy of the height measurement is greatly improved compared with the digital image (Fig. 11). Therefore, compared with digital images, the use of the LiDAR point cloud data leads to accurate estimates of plant height.

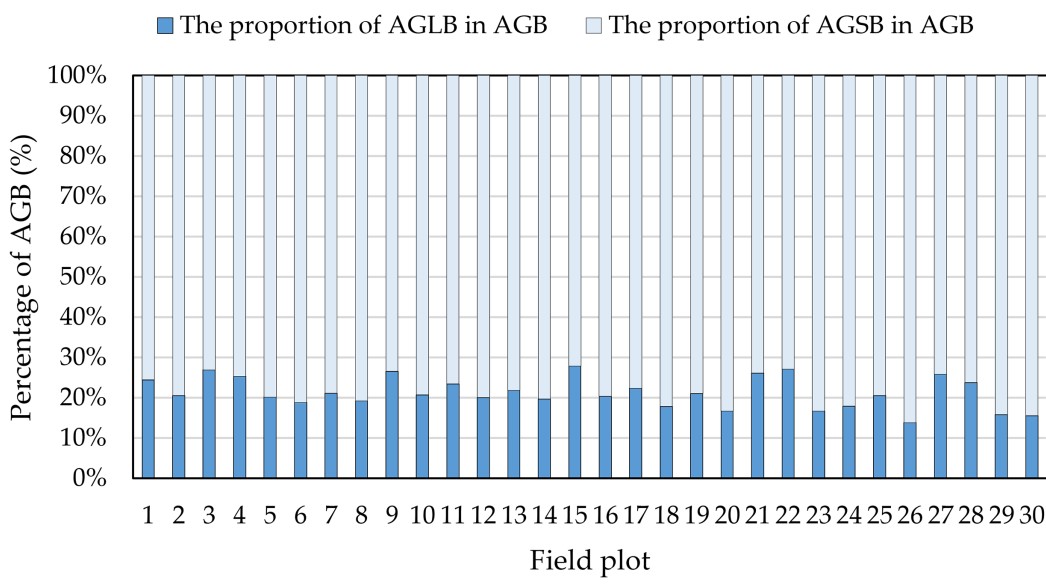

**Figure 12 Proportion of AGLB and AGSB in AGB.**

Although many studies have estimated crop biomass based on airborne spectral data (*Bendig et al., 2015*; *Fu et al., 2014*; *Gnyp et al., 2014*; *Kross et al., 2015*; *Liu et al., 2010*), multispectral data have only limited ability to estimate crop biomass. In fact, because airborne multispectral images only observe the upper canopy leaves, the structure below the canopy is not monitored when the vegetation density is high, resulting in limited information on the vertical structure of the observed vegetation. Moreover, when the crop density in the observation area is large, the spectral signal from the spectral sensor saturates (*Baret & Guyot, 1991*; *Turner et al., 1999*). In addition, we find that the percent of AGLB and AGSB in the AGB is 13.77–27.84% and 72.16–86.23%, respectively (Fig. 12). These results show that the AGLB accounts for no more than 30% of the AGB. Therefore, since it is difficult to obtain the vertical structure information of the crop from the spectral data, estimating the AGB of the crop based on the spectral data alone may result in less accurate estimates of crop biomass (*Wang et al., 2017*).

In addition, from the one-way ANOVA, since the *p*-value is far less than 0.05, it can be inferred that, when the plant density differs between plots, the mean value of the AGB differs significantly. Second, based on the multiple comparison results and the homogeneity of the variance, we conclude that the mean difference in AGB is statistically significant when the *p*-value is less than 0.05, when the density F (1–10) is compared with the density G (11–20), the density F is compared with the density H (21–30), the density F is compared with the density I (>30), and the density G is compared with the density I. However, when the density H is compared with the density G and the density H is compared with the density I, the mean difference in AGB is not statistically significant. This may increase the density of the plants in the given plot, and plant height also increases due to competitive-growth effects between plants, but the overall biomass of the plot does not differ significantly from the biomass of the density G and I, resulting in no statistical significance between density H and G, I (Table 6).

**Table 6 Multiple comparison result for one-way ANOVA.**

| Density (I) | Density (J) | Mean difference (I–J) | Standard error | p-value | 95% Confidence interval | |
|---|---|---|---|---|---|---|
| | | | | | Lower limit | Upper limit |
| F | G | −242.22 | 55.37 | 0.000 | −356.05 | −128.40 |
| | H | −318.02 | 57.07 | 0.000 | −435.34 | −200.69 |
| | I | −420.17 | 70.58 | 0.000 | −565.26 | −275.07 |
| G | F | 242.22 | 55.37 | 0.000 | 128.40 | 356.05 |
| | H | −75.79 | 57.07 | 0.196 | −193.11 | 41.53 |
| | I | −177.94 | 70.58 | 0.018 | −323.03 | −32.84 |
| H | F | 318.02 | 57.07 | 0.000 | 200.69 | 435.34 |
| | G | 75.79 | 57.07 | 0.196 | −41.53 | 193.11 |
| | I | −102.15 | 71.93 | 0.167 | −250.00 | 45.70 |
| I | F | 420.17 | 70.58 | 0.000 | 275.07 | 565.26 |
| | G | 177.94 | 70.58 | 0.018 | 32.84 | 323.03 |
| | H | 102.15 | 71.93 | 0.167 | −45.70 | 250.00 |

**Note:**
In the table, when the p-value is less than 0.05, the mean difference in AGB is statistically significant.

## CONCLUSIONS

Remote sensing UAV platforms equipped with LiDAR and multispectral sensors offer the advantages of flexible operation, convenient data acquisition, and high-spatial resolution. In this study, we use the LiDAR and multispectral data acquired from the UAV platform to evaluate their use for estimating the AGB of maize. This paper proposes the stem-leaf separation strategy integrated with UAV LiDAR and multispectral image data to estimate the AGB of maize.

According to the screened indicators, the SVIs derived from multispectral data and the LSPs derived from LiDAR data were subjected to MLR with the AGLB and AGSB, respectively. At the same time, all SVIs derived from multispectral data and all LSPs derived from LiDAR data were subjected to PLSR with the AGLB and AGSB, respectively. Next, the estimated values of AGLB and AGSB were added to a single estimation for AGB. The results indicate a strong correlation between the estimated and field-observed of AGB using the MLR method ($R^2 = 0.82$, RMSE = 79.80 g/m², NRMSE = 11.12%) and the PLSR method ($R^2 = 0.86$, RMSE = 72.28 g/m², NRMSE = 10.07%). The results show that PLSR is more accurate for estimating AGB than MLR, with $R^2$ increasing by 0.04, RMSE decreasing by 7.52 g/m², and NRMSE decreasing by 1.05%. In addition, the AGB is more accurately estimated by combining LiDAR with multispectral data than LiDAR and multispectral data alone, with $R^2$ increasing by 0.13 and 0.30, respectively, RMSE decreasing by 22.89 and 54.92 g/m², respectively, and NRMSE decreasing by 4.46% and 7.65%, respectively. This result reflects a significant improvement in the accuracy of the estimated AGB of maize.

Thus, the findings of this study lead us to conclude that this technology would allow for the convenient surveillance of maize to observe growth trends and could therefore provide guidance in agriculture management decisions. Although the statistical tests showed the

effectiveness of the stem-leaf separation strategy, the scale of maize plants tested in this study is relatively small. Out next study will therefore assess different scales of maize planting to determine the effectiveness of the proposed method in real production scenarios.

## ACKNOWLEDGEMENTS

Thanks to Weiguo Li, Hong Chang, Dong Han, Chuanlong Ding, Yulong Wang, Bo Zhang, Rui Chen, and Zhuangzhi Yuan for the field data collection and farmland management. Thanks to Hao Yang, Liang Han, and Zhenhai Li for their help in my research and writing of this paper. We are grateful to the anonymous reviewers for their helpful comments and suggestions on the manuscript.

### Funding

This work was supported by the National Key Research and Development Program of China (2017YFE0122500), the Natural Science Foundation of China (41771469 and 41571323), the Beijing Natural Science Foundation (6182011), the Beijing Academy of Agriculture and Forestry Sciences (KJCX20170423), and the Special Funds for Technology innovation capacity platform building sponsored by the Beijing Academy of Agriculture and Forestry Sciences (PT2019-29). The funders had no role in study design, data collection and analysis, decision to publish, or preparation of the manuscript.

### Grant Disclosures

The following grant information was disclosed by the authors:
National Key Research and Development Program of China: 2017YFE0122500.
Natural Science Foundation of China: 41771469 and 41571323.
Beijing Natural Science Foundation: 6182011.
Beijing Academy of Agriculture and Forestry Sciences: KJCX20170423.
Special Funds for Technology innovation capacity platform building sponsored by the Beijing Academy of Agriculture and Forestry Sciences: PT2019-29.

### Competing Interests

The authors declare that they have no competing interests.

### Author Contributions

- Yaohui Zhu conceived and designed the experiments, performed the experiments, analyzed the data, contributed reagents/materials/analysis tools, prepared figures and/or tables, authored or reviewed drafts of the paper, approved the final draft.
- Chunjiang Zhao approved the final draft.
- Hao Yang performed the experiments, contributed reagents/materials/analysis tools, authored or reviewed drafts of the paper, approved the final draft.
- Guijun Yang authored or reviewed drafts of the paper.
- Liang Han contributed reagents/materials/analysis tools.
- Zhenhai Li contributed reagents/materials/analysis tools.
- Haikuan Feng performed the experiments.
- Bo Xu performed the experiments.
- Jintao Wu performed the experiments, prepared figures and/or tables.
- Lei Lei performed the experiments, prepared figures and/or tables.

## Data Availability

Multiple linear regression model (Matlab) and partial least squares regression model (PLSR) are available as Supplemental Files.

## Supplemental Information

Supplemental information for this article can be found online at http://dx.doi.org/10.7717/peerj.7593#supplemental-information.

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
