# Peer review of "Estimation of maize above-ground biomass based on stem-leaf separation strategy integrated with LiDAR and optical remote sensing data"

_PeerJ, doi:10.7717/peerj.7593_

## Round 0.1 · original submission · Major Revisions

The paper has been evaluated by 2 reviewers who suggested major revision.

Substantial Improvements need to be made to the writing, some citations were misrepresented, other areas needed citations, and more detail in the methodology is required to ensure reproducibility of this research.

[]

·

Basic reporting

1) I noticed a few citations which follow statements that do not accurately reflect the findings of what is being cited. The two most noticeable examples of this are Line 108 with James and Robinson 2014 (Lidar is not mentioned in this paper, it focuses on Structure from Motion) and line 363 with Schlemmer et al. 2013 (AGB is not mentioned in this paper, and the paper concludes that nitrogen and chlorophyll content can be accurately measured using only multispectral data). I recommend you review your citations and ensure that the way they are being presented in your paper is accurate.

2) Several statements were made which should have been followed by citations backing up the claim. Examples can be found in lines 75, 117, 233-234, and 248-249.

3) The English language should be improved to ensure that an international audience will fully comprehend your work. Examples of areas which could use improvement can be found in lines 96-99, 147-148, 210, and 240.

4) As some readers, myself included, may not be familiar with NRMSE I recommend providing a citation which provides an explanation for the approach and how to interpret the results in line 249.

Experimental design

The paragraph from lines 227-237 needs more detail in order to make your experiment methodology clear and reproducible. I recommend providing more detail on which metrics were used, why you selected the method used here (citations if based on other studies), and how the SVI and LSP data were fused.

Validity of the findings

As this study generates a point cloud, results for the generated point cloud should be provided. These include the average point spacing, point density, and difference in Lidar derived elevation when compared to elevation values gathered via GPS. Most of this information is present throughout your paper but it would be more accessible if organized in a subsection of your results.

Additional comments

1) I would recommend stating the SVIs/LSPs used in the model instead of the more general SVI/LSP language used in the methods/results/discussion sections.

2) The paragraph from lines 339-352 discusses an aspect of this study which is not clearly outlined earlier in the methodology and results. I recommend adding a subsection to both sections in order to (1) outline what was done to generate the image-based elevations and (2) present the results of this process. Otherwise, this paragraph seems out of place as it does not relate to what has been discussed so far.

3) The paragraph from lines 328-337 should also be discussed in the methodology. The results are appropriate here, but the steps taken to get these results need to be outlined as part of your methodology.

4) The paragraph from lines 364-376 should have a corresponding paragraph in the results section outlining the ANOVA results.

5) Overall, I though your research was interesting and your approach novel. I think your paper can best be improved by double checking citations to ensure they are cited correctly and where needed; ensuring that outcomes discussed have a corresponding component in the methodology/results sections; improve language clarity in some areas; and providing more detail to the previously noted paragraph in your methodology.

Reviewer 2 ·

Basic reporting

This paper described how to estimate maize above-ground biomass based on stem-leaf separation strategy integrated with LiDAR and optical remote sensing data. This is an interesting topic, dealing with a still unresolved methodology suitable for the different conditions of the plants at a global scale: to get AGB from drone remote sensing. The writing is clear. Overall, the manuscript is publishable, but, needs a critical revision first. There are some minor grammatical and structural mistakes in the manuscript and I am not able to highlight/revise all of them. So, please revise the manuscript and improve it considerably in terms of grammatical and structural mistakes.
The literature is suitable for the purpose of the article.

Experimental design

1) Within aims and scope of the PeerJ journal.
2) Research question well defined, relevant & meaningful. It is stated how the research fills an identified knowledge gap.
3) Structure conforms to technical & ethical standard.
4) Methods described with sufficient detail & information to replicate.

Validity of the findings

1) Data is robust, statistically sound, & controlled.
2) Conclusions are well stated, linked to original research question & limited to supporting results.
3) Speculation is welcome.

Additional comments

1. The research is well defined and relevant although the research gap was vaguely presented (possibly as a result of weak command of the English language). The authors should modify the introduction.
2. Line 151: Did the authors consider different growth cycles? It may result in some uncertainty. The authors should consider different periods to reduce the uncertainty and improve the accuracy of the predictions.
3. Line 149: The sample size should be added in the context, more than the site map. In addition, a total 30 samples are relatively small, which used to construct models. Did the authors divide them into calibration set and validation set? These contents should be clear.
4. Line 210: Why did select these spectral indices? Please add the declarations.
5. Why the RPD (ratio of performance to deviation) and/or RPIQ (ratio of performance to inter-quartile distance) was not used in this study? In common, these parameters are more effective than R2 and NRMSE.
6. The authors should redesign Table 1, the description (full name of indices) and references should be separated.
7. There is no need for Table 3. The involved content is common knowledge.
8. In all the numbers with decimals, please consider to homogenize to only two decimals.
9. better to mention how large is the plot is.
10. I would suggest that the authors more clearly describe the processing of drone images, in particular, the parameters that have been used in the atmospheric correction such as the atmospheric mode and aerosol model that should be mentioned.
11. Reflectance spectra curves of soils with different AGB should be added in the revision.
12. The clear descriptions of spatial resolutions of UAV Multispectral Data and LiDAR data should be included. How do you deal with this specific issue, were they resampled?
13. I recommend the authors add a clear flight path in this study.
14. Why MLR was used in this study? In fact, this method is too old. PLSR is a commonly used approach to build a multi-variable model, with the advantages of dimension reduction and handling of collinearities among various independent variables. PLSR is better than MLR. In addition, the application of MLR has a strict requirement (normal distribution).
15. The detailed threshold of p-value of 0.01 and 0.05 should be added.
16. The introduction of confidence level is welcome, however, they should be displayed in the Figure. (Refer to Quantitative estimation of soil salinity by means of different modeling methods and visible-near infrared (VIS–NIR) spectroscopy, Ebinur Lake Wetland, Northwest China[J]. PeerJ, 2018, 6: e4703.)
17. The pretreatments of drone data could improve the sensitivity between the dependent and independent variables in the spectral analysis. The authors should consider them.
18. At present, the section of discussion is too weak. The relevant contents of estimation mechanism, introduction of machine learning algorithms, comparisons between of different models… are welcome.

---

## Round 0.2 · Minor Revisions

The paper has been revised accordingly. However there are still few snags. In particular, the English needs much improvement. As this is an international journal, a thorough English editing is required to ensure that readers can easily understand the article. Note that English improvement is a requirement for this paper to be further considered.

Few technical issues:
- R2 formula is wromg, should be 1 - RSS/TSS. (Missing 1 -)
- RPD and R2 are the same (R2 uses variance, while RPD uses standard deviation) Thus I suggest to remove RPD measure
- Please explain how many number of factors used in the PLSR model and how it was determined.

Reviewer 2 ·

Basic reporting

As I can see from the previous review, you have corrected all of the reviwers comments. It is ready for publication.

Experimental design

no

Validity of the findings

no

---

## Round 0.3 · accepted · Accept

The paper has been revised accordingly and ready for publication.